# Polymorphism of feldspars above 10 GPa

Anna Pakhomova [1✉], Dariia Simonova[2], Iuliia Koemets[2], Egor Koemets[2], Georgios Aprilis [3],
Maxim Bykov [2], Liudmila Gorelova [4], Timofey Fedotenko[3], Vitali Prakapenka [5] & Leonid Dubrovinsky [2]

Feldspars are rock-forming minerals that make up most of the Earth's crust. Along the mantle geotherm, feldspars are stable at pressures up to 3 GPa and may persist metastably at higher pressures under cold conditions. Previous structural studies of feldspars are limited to ~10 GPa, and have shown that the dominant mechanism of pressure-induced deformation is the tilting of $AlO_4$ and $SiO_4$ tetrahedra in a tetrahedral framework. Herein, based on results of in situ single-crystal X-ray diffraction studies up to 27 GPa, we report the discovery of new high-pressure polymorphs of the feldspars anorthite ($CaSi_2Al_2O_8$), albite ($NaAlSi_3O_8$), and microcline ($KAlSi_3O_8$). The phase transitions are induced by severe tetrahedral distortions, resulting in an increase in the Al and/or Si coordination number. High-pressure phases derived from feldspars could persist at depths corresponding to the Earth upper mantle and could possibly influence the dynamics and fate of cold subducting slabs.

[1] Deutsches Elektronen-Synchrotron (DESY), 22607 Hamburg, Germany. [2] Bayerisches Geoinstitut, Universität Bayreuth, 95440 Bayreuth, Germany. [3] Material Physics and Technology at Extreme Conditions, Laboratory of Crystallography, Universität Bayreuth, 95440 Bayreuth, Germany. [4] Institute of Earth Sciences, Saint-Petersburg State University, Saint-Petersburg, Russia 199155. [5] Center for Advanced Radiation Sources, University of Chicago, Chicago, IL 60637, USA. ✉email: anna.pakhomova@desy.de

Albite ($NaAlSi_3O_8$), anorthite ($CaAl_2Si_2O_8$) and K-feldspar (microcline, sanidine, orthoclase; $KAlSi_3O_8$) are the major feldspar minerals that are abundant in various geological environments. Feldspars make up ~50–60% of the total volume of the Earth's crust, composing many types of igneous[1], meta-morphic[2] and sedimentary[3] rocks. Feldspars of the anorthite-albite series (i.e., plagioclases) are the most common on the Earth's surface and are widely distributed on other planetary bodies of the inner Solar System. Plagioclases compose most of the Moon's crust and have been detected on the surfaces of Mars[4,5], Venus[6] and Mercury[7], as well as in chondrites[8]. Such high geological relevance of feldspars has led to numerous experimental and theoretical studies on high-pressure (P)-high-temperature (T) phase relations[9–11], elastic[12,13] and rheological properties[14], and the amorphization mechanism[15].

Under ambient conditions, the three-dimensional framework of feldspars is based on $TO_4$ tetrahedra (T=$Si^{4+}$, $Al^{3+}$), where low-charge cations ($K^+$, $Na^+$, $Ca^{2+}$) occupy the large voids (Fig. 1). Commonly, feldspars are thermodynamically stable at pressures up to ~3 GPa along the normal mantle geotherm[9–11]. At room-temperature, the decomposition reactions of feldspars are kinetically hindered, allowing investigations of metastable feldspars at higher pressures. All previous structural studies on feldspars were performed below 10 GPa and agree that the pressure-induced compression of the framework is primarily accommodated by altering the T–O–T bond angles[13,16–18], while the $TO_4$ tetrahedra show very little compression and behave as relatively rigid units.

Herein, we report on the discovery of new high-pressure polymorphs of anorthite, albite and microcline at pressures above 10 GPa. In contrast to the previous observations, the pressure-induced transitions are induced by severe geometrical distortion of the $TO_4$ tetrahedra, resulting in an increase in the Al and/or Si coordination number and formation of denser frameworks based on $TO_4$, $TO_5$ and $TO_6$ structural units. Pressure–temperature (P–T) existence fields of the new feldspar phases indicate that these materials can persist at greater depths than previously thought along colder than average mantle geotherms. We propose that dense feldspar phases can withstand deep subduction along (ultra)cold subduction zones and influence the dynamics and fate of descending slabs.

## Results

**High-pressure anorthite-III with Al[4 + 1], Al[5] and Al[6].** Single-crystal X-ray diffraction experiments allowed us to follow the high-pressure evolution of the anorthite crystal structure at pressures of up to ~20 GPa (Supplementary Table 1). Above 2.2 GPa, anorthite ($CaSi_2Al_2O_8$) undergoes a displacive first-order phase transition to the $I$-1 phase, in agreement with previous observations of Angel (1992) up to 5 GPa, who bracketed the transition pressure between 2.55 and 2.74 GPa[19] (Supplementary Fig. 1). Across the anorthite→anorthite-II transition, the tetra-hedral feldspar framework is preserved. Our results show that the $I$-1 phase of anorthite is stable at least up to 8.8 GPa, while it transforms to a new high-pressure polymorph between ~9 and 11 GPa. The crystal structure of anorthite-III at 11.1 GPa was solved and refined in the $P$-1 space group (Supplementary Table 2). The phase transition is accompanied by drastic mod-ifications of the tetrahedral framework, mostly pronounced in an increase in the Al coordination number. The structurally distinct Al1, Al2 and Al3 sites retain their distorted tetrahedral coordi-nation, while Al5 and Al6 are surrounded by five oxygen atoms and form $AlO_5$ polyhedra with trigonal bipyramidal geometry (Supplementary Table 3). The same geometry can be considered for the Al4 atom; however, this atom is surrounded by four oxygens within 1.72–1.78 Å and one oxygen at a distance of 2.2 Å, indicating that at this pressure point, the polyhedral coordination can likely be better presented as [4 + 1]. Al7 and Al8 form $AlO_6$ octahedra and are coordinated to six neighbouring atoms. The average Al–O bond lengths for $AlO_4$, $AlO_5$ and $AlO_6$ polyhedra are 1.735, 1.829 and 1.922 Å, respectively. All Si atoms preserve tetrahedral coordination during the transition. Upon formation of a three-dimensional framework, the $Al7O_6$ octahedra and $Si8O_4$ tetrahedra share common edges, while the other $AlO_4$, $AlO_5$, $AlO_6$ and $SiO_4$ polyhedra share vertices (Fig. 2a). Ca atoms, located in the framework voids, are eightfold (Ca2, Ca3) and ninefold (Ca1, Ca4) coordinated (taking into account Ca–O distances <3 Å). The anorthite-III crystalline phase is stable at least up to 16 GPa. Between 16 and 22 GPa, it undergoes amor-phization, as indicated by the absense of single-crystal reflections in a diffraction pattern collected at 22 GPa.

**High-pressure albite-II and albite-III.** Our observations on the compression behaviour of albite ($NaAlSi_3O_8$, sp.gr. $C$-1) are in agreement with previous single-crystal X-ray diffraction studies performed up to 9.4 GPa[20,21] (Supplementary Table 1 and Sup-plementary Fig. 1). According to our results, stability field of $C$-1 phase extends up to 11.5 GPa while between 11.5 and 13.5 GPa, it transforms to a new phase, albite-II. The crystal structure of the albite-II was solved and refined in the $P$-1 space group at 13.5 GPa (Supplementary Table 2). In the crystal structure of albite-II, Si sites retain tetrahedral coordination, while the only Al site gains two more oxygen atoms in its coordination sphere, thus increasing its coordination number to six (Fig. 2b). Al atoms have distorted octahedral coordination, with an average bond length of

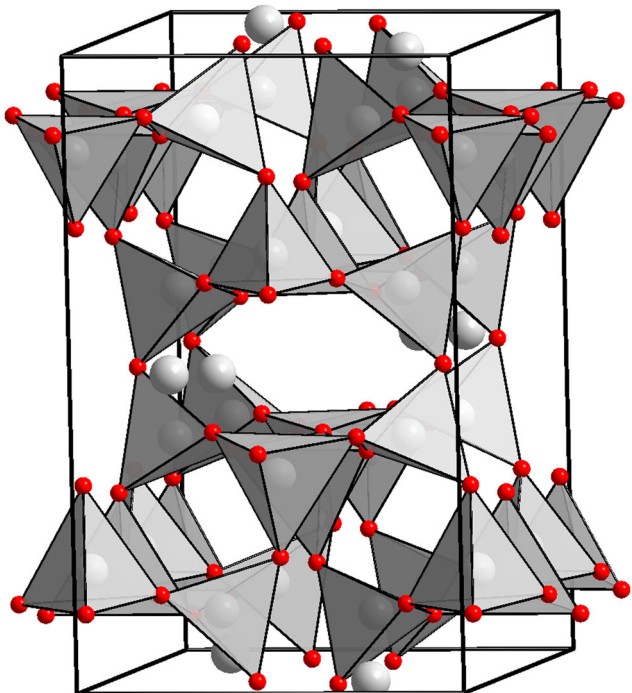

**Fig. 1 Feldspar crystal structure under ambient conditions.** All Si and Al atoms are bonded to four oxygen atoms to form tetrahedra. $SiO_4$ and $AlO_4$ tetrahedra (given in grey) form a three-dimensional framework by sharing common vertices. Al atoms occupy half of the tetrahedral sites in anorthite ($CaSi_2Al_2O_8$) and a quarter of the sites in albite ($NaAlSi_3O_8$) and microcline ($KAlSi_3O_8$). Large cations ($Ca^{2+}$, $Na^+$, $K^+$) located in the framework voids are represented grey spheres. Oxygen atoms are given in red. Black lines outline the unit cell of the aristotype structure.

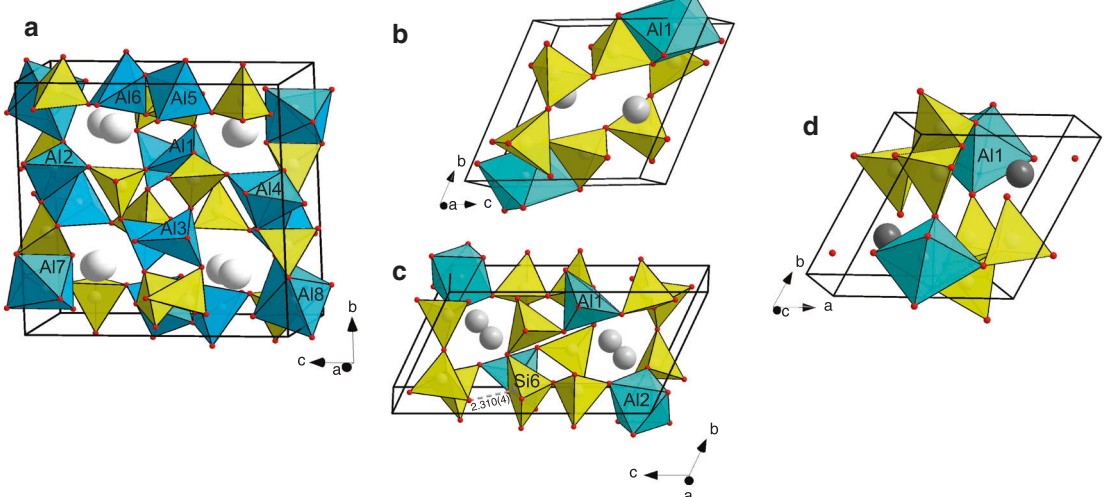

**Fig. 2 Crystal structures of the discovered high-pressure polymorphs of feldspars.** $SiO_n$ and $AlO_n$ ($n = 4$–6) polyhedra are given in yellow and blue, respectively. O, Ca, Na and K atoms are presented as red, light-grey, grey and dark-grey spheres, respectively. Black lines outline unit cells. **a** Anorthite-III at 11.1 GPa. Al1, Al2 and Al3 atoms, as well as all silicon atoms retain tetrahedral coordination typical for feldspars. Al4, Al5 and Al6 have trigonal-bipyramidal coordination, while Al7 and Al8 are octahedrally coordinated. **b** Albite-II at 13.5 GPa. The Al1 atom is in octahedral coordination, while all silicon atoms are tetrahedrally coordinated. **c** Albite-III at 17.5 GPa. Al1 and Al2 have five- and six-fold coordination, respectively. All silicon atoms except Si6 are tetrahedrally coordinated. Si6 is in square pyramidal coordination, and a decrease in the Si6–O10 contact (marked as a dashed grey line) indicates a progressive change from polyhedral geometry to octahedral. **d** Microcline-II at 12.8 GPa. The Al1 atom has octahedral coordination, while all silicon atoms are tetrahedrally coordinated.

1.850 Å (Supplementary Table 4). Two $AlO_6$ octahedra share a common edge, forming a dimer. The dimers are connected to $SiO_4$ tetrahedra by sharing common vertices upon formation of a three-dimensional framework. Na atoms are eightfold coordinated with an average <Na–O> bond length of 2.412 Å. Between 13.5 and 16 GPa, albite-II undergoes one more displacive first-order phase transformation accompanied by a further increase in the coordination number of Si atoms (Fig. 2c). The crystal structure of the albite-III was solved and refined in the $P$-1 space group at 17.5 GPa (Supplementary Table 2). In the crystal structure of albite-III, five of six Si sites remain tetrahedrally coordinated (Supplementary Table 5). The sixth Si6 site is coordinated to five oxygens, forming a square pyramid with an average Si–O bond length of 1.711 Å. However, the strong shift of Si6 from the centre of the square pyramid towards the square base as well as the presence of a sixth point of contact (Si6–O10) of 2.31(1) Å indicate the tendency of the Si6 atom to form an octahedron. Such a bond distance distribution suggests that the polyhedral geometry should be better described as [5 + 1]. Upon further compression to 20 GPa, the Si6–O10 bond distance decreases quickly to 2.113(4) Å, showing that the O10 atom continuously approaches the Si6 coordination sphere. One structurally distinct site, Al1, is coordinated by five oxygen atoms, forming an $AlO_5$ trigonal bipyramid. Different geometry is observed for the Al2 site: it is coordinated by six oxygen atoms, forming an $AlO_6$ octahedron. The average bond distances for $AlO_5$ and $AlO_6$ polyhedra are 1.808 and 1.859 Å, respectively. $TO_n$ (T=Al, Si; $n = 4$–6) polyhedra share common vertices and edges to form a three-dimensional framework. Na atoms are sevenfold coordinated with an average <Na–O> bond distance of 2.35 Å (taking into account a coordination sphere of 3 Å). The crystal structure of albite-III is stable at least up to the highest studied pressure of 20 GPa. Upon full decompression, the albite-III transforms to the initial $C$-1 phase (albite) based on the tetrahedral framework. Notably, the albite-II phase was observed upon decompression, while upon compression, the range between 11.5 and 16 GPa was missed (Supplementary Table 1).

**High-pressure microcline-II with Al[6].** At pressures up to 7 GPa, the tetrahedral framework of microcline ($KAlSi_3O_8$, sp.gr. $C$-1) undergoes continuous anisotropic compression, in agreement with previous observations[22] (Supplementary Table 1 and Supplementary Fig. 1). Our new data indicates that at elevated pressures between 7 and 9 GPa, the $C$-1 phase starts to experience elastic softening, as indicated by behaviour of the unit cell parameters: increase of the $a$ axis and pronounced decrease of the $b$ axes. While the initial tetrahedral structure is preserved, the softening of the microcline continues up to 10.5 GPa. Unfortunately, the only three available data points between 9 and 10.5 GPa are insufficient for the quantitative estimation of the elastic softening of microcline in this pressure range. Upon further compression, between 10 and 13 GPa, the $C$-1 phase undergoes a first-order displacive phase transition induced by an increase in the Al coordination number from four to six (Fig. 2d). At 12.8 GPa, $AlO_6$ octahedra possess a distorted geometry, with Al–O bonds varying from 1.76–2.16 Å and an average bong length of 1.89 Å (Supplementary Table 6). Throughout the transition, $SiO_4$ tetrahedra are preserved. Two $AlO_6$ octahedra share a common edge, forming a dimer. The dimers are connected to $SiO_4$ tetrahedra by sharing common vertices upon formation of a three-dimensional framework. Potassium atoms are ninefold coordinated with an average <K–O> bond length of 2.664 Å. Microcline-II is stable at least up to the highest studied pressure of ~27 GPa.

**New high-pressure feldspar polymorphs: mechanism of coordination number increase.** Previous high-pressure structural studies on feldspars up to ~10 GPa repeatedly indicated that the framework mainly responds to pressurisation by rotation of $TO_4$ tetrahedra and changes in T–O–T angles while the tetrahedra undergo little distortion and compression[16–18]. In contrast, we have discovered that compression of the feldspar framework at higher pressures is governed by a different mechanism, i.e., by severe geometrical distortion of tetrahedra. Above ~10 GPa, the $TO_4$ tetrahedra do not behave as nearly rigid units but transform

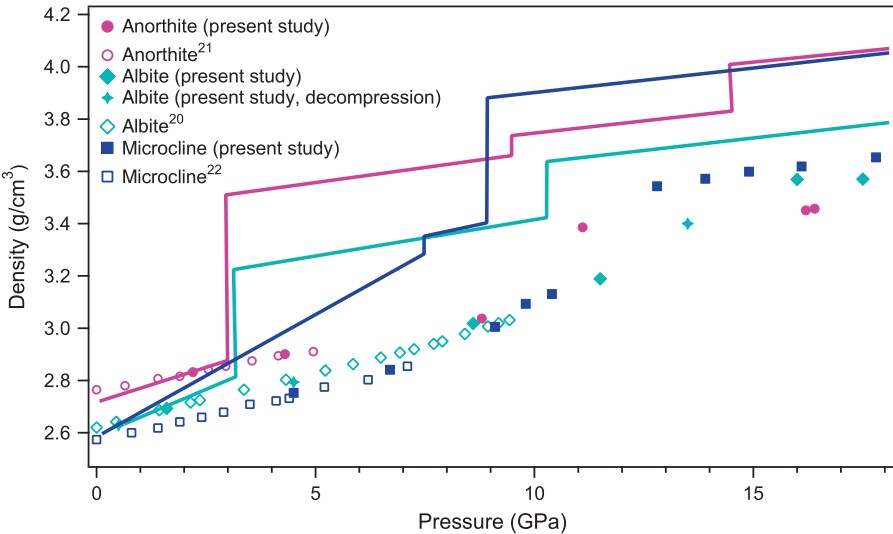

**Fig. 3 Density-pressure evolution of feldspar compositions.** Densities of metastable feldspars obtained in room-temperature, high-pressure single-crystal X-ray diffraction experiments are given as filled (present study) and open markers[20-22]. Above ~11 GPa, densification of the framework is realised through an increase in Al or/and Si coordination number to five- and/or six-fold. Solid lines represent the evolution of feldspar compositions according to their decomposition reactions along P–T conditions of the normal mantle geotherm[34].

into $TO_5$ and $TO_6$ polyhedra by forming additional T–O bonds. As the ionic potential (defined as the ratio of ionic charge to ionic radius[23]) is lower for $Al^{3+}$ ($3^+/0.39$ Å $= 7.7$) than for $Si^{4+}$ ($4^+/0.26$ Å $= 15.4$), the Al-centred tetrahedra are more compressible than $SiO_4$. This explains why, during framework densification, only $AlO_4$ tetrahedra undergo geometrical distortion (i.e., for anorthite and microcline) or undergo the transition first (i.e., for albite). Along the sequence $TO_4 \rightarrow TO_5 \rightarrow TO_6$, the average T–O distances in a polyhedron increase (for instance, compare the distances of 1.735, 1.829 and 1.922 Å for $AlO_4$, $AlO_5$ and $AlO_6$, respectively, in anorthite-III) so that the newly formed polyhedra are more compressible due to the formation of longer T–O bonds. This allows the feldspar crystal structure to further adapt to high-pressure conditions by pronounced compression of the soft $TO_5$ and $TO_6$ units. Notably, the distribution of Si–O and Al–O bond distances in the studied feldspars does not provide evidence for Si-Al disorder across the T sites.[24]

Pressure-induced phase transformations accompanied by coordination number increase of T atoms are preceded by progressive geometrical distortion of $AlO_4$ units in the initial tetrahedral structures of feldspars. Thus, for the $C\bar{1}$ phase of microcline, deviation of the $AlO_4$ units from the ideal tetrahedral geometry above ~7 GPa is clearly visible on plots showing the bond angular variation (BAV) and quadratic elongation (QE) parameters as a function of pressure (Supplementary Fig. 2b, c)[25]. As shown in Supplementary Fig. 2a, the distortion originates from the closure of six-membered rings and progressive approach of the additional O3* oxygen into the coordination sphere of Al atom across the ring. Such structural response results in significant elastic softening in the microcline in the pressure range of ~9–10.5 (Supplementary Fig. 1), preceding formation of the new phase observed at 12.8 GPa. The elastic softening recently discussed for albite[13,20,26] has also been proposed to indicate the upcoming phase transition[20], that has been indeed observed in the present study. Regarding anorthite, the only available data point at 8.8 GPa between ~4 and 11 GPa (when anorthite-III is already formed) is insufficient to draw conclusions about pre-transitional softening of the anorthite-II crystal structure. However, increases in the distortion parameters of the $AlO_4$ tetrahedra (BAV up to ~128° and QE up to ~1.04) indicate that

anorthite-II $\rightarrow$ anorthite-III phase transition is also induced by continuous distortion of $AlO_4$ tetrahedra. A similar mechanism of pressure-induced densification has recently been observed for tetrahedral framework compounds with different connectivities of $TO_4$ units (T=$Si^{4+}$, $Al^{3+}$, $P^{5+}$, $Be^{2+}$)[27-29]. Similar to feldspars, the displacive $TO_4 \rightarrow TO_5$ transitions are induced by continuous approach of an additional O atom to the coordination spheres of T atoms and are accompanied by severe geometrical distortion of initial $TO_4$ tetrahedra. In this type of compounds, the newly formed $TO_5$ polyhedra are of either trigonal-bipyramidal or of square pyramidal geometries. However, it is likely that square pyramidal geometry is a transitional step in the formation of octahedra, as indicated by the progressive approach of a sixth oxygen in albite-III. Meanwhile, the newly $TO_6$ polyhedra are exclusively of octahedral geometry.

For most of the studied frameworks[27-29], the formation of an additional T–O bond is accompanied by a pronounced jump in the unit cell parameters, as also found in the current study and, accordingly, an increase in the densities of the feldspar-derived phases (Fig. 3). By extrapolation of previously reported equations of state[20-22], we can note ~8.5%, 12% and 15% differences between the expected densities of the initial tetrahedral framework and that of a framework based on $TO_n$ units ($n = 4–6$) (for anorthite-III at 11.1 GPa, albite-III at 16 GPa and microcline-II at 12.8 GPa, respectively).

**High-temperature stability of the discovered high-pressure phases.** A series of ex situ multi-anvil experiments (Supplementary Table 7) were performed to test whether the phases of albite and anorthite observed at high-pressure and room-temperature are preserved upon heating. At all studied P–T points (12 GPa and 800 °C; 13 GPa and 800 °C; 17 GPa and 700 °C), albite decomposes upon heating into an assembly of jadeite and stishovite, in agreement with previous reports on the phase stability and decomposition reactions of this material[9,30]. Surprisingly, a material with an anorthite composition withstands heating up to 500 °C at 13 GPa and up to 600 °C at 14 and 15 GPa (Fig. 4). An additional experiment under wet conditions at 5 GPa and 500 °C with longer heating time of 110 h has also demonstrated the stability of anorthite.

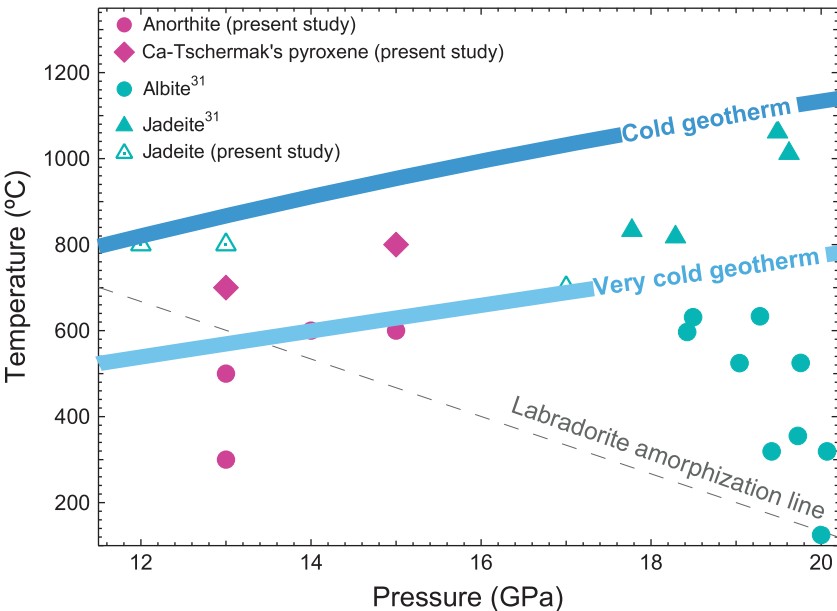

**Fig. 4 Phase relations in feldspars of the anorthite-albite series as revealed by multi-anvil experiments in the present study and in the work of Kubo et al.[30].** Cold slab and very cold slab geotherms are from Zhang et al.[41]. The dashed grey line represents the amorphization line of labradorite with a composition of $Ab_{45.0}An_{51.8}Or_{3.2}$[30].

In-house and synchrotron-based single-crystal diffraction experiments (SCXRD) of the recovered crystals under ambient conditions shows that they have an initial anorthite tetrahedral framework structure. Recently, Kubo et al.[30] has shown that labradorite (intermediate composition of 51.8% anorthite, 45.0% albite and 3.2% microcline) and albite remain in a crystalline state up to 13.0 GPa at 660 °C and up to 26.3 GPa at 950 °C, respectively[30]. Our SCXRD results indicate that all crystalline phases studied by Kubo et al.[30] over a wide P–T range (Fig. 4) may be high-pressure polymorphs of feldspars featuring five- and six-coordinate Al and/or Si. The new phases cannot be identified by energy-dispersive XRD, and further experiments on labradorite are needed to clarify which phases Kubo et al.[30] actually observed.

Upon heating to higher temperatures (700 °C at 13 GPa; 800 °C and 1500 °C at 15 GPa), the anorthite decomposes into Al- and Ca-rich clinopyroxene, so-called calcium Tschermak's pyroxene[31,32]. At temperatures of 700 °C at 13 GPa and 1500 °C at 15 GPa, the quenched product was composed of large crystallites, allowing the identification and characterisation of phases by SCXRD. On the basis of the crystal structure refinement, the chemical formula of the pyroxene was determined to be $(Al_{0.47(1)}Ca_{0.53(1)})Ca(Si_{1.6}Al_{0.4})O_6$ (Supplementary Table 8). In contrast, two experiments at 800 °C and 15 GPa produced fine-grained polycrystalline pieces co-existing with transparent amorphous pieces. The fitting of the powder patterns revealed the presence of grossular in addition to pyroxene. The transparent pieces are assumed to be amorphized anorthite, which agrees with previous multi-anvil experiments on feldspars[30]. Our results contradict the laser-heating experiment of Gautron & Madon[33], where the stability of anorthite at temperatures of up to ~1800 °C in the pressure range of 12.5–17 GPa was reported. Note that these observations have been repeatedly questioned[16,34].

Another recent ex situ multi-anvil study on pure anorthite composition focused on P–T conditions of 1400–2400 °C and 14–25 GPa[34]. These experiments demonstrated the formation of phase assemblages of grossular (Gr) + kyanite (Ky) + stishovite (St) at ~14 GPa and Gr + calcium-alumino-silicate phase (CAS)

+ St at ~18 GPa. The same assemblage of Gr + CAS + St was recently detected upon decomposition of lunar highland anorthosite after heating to 1700 °C in a multi-anvil press in the pressure range of 12–22 GPa[35]. In contrast to our results, Tschermak's pyroxene was not observed in either multi-anvil study[34,35]. Most of our experiments were performed at temperatures below 800 °C (Supplementary Table 7), but in the one high-temperature experiment at 15 GPa and 1500 °C, we also identified pyroxene as a recovered product. This discrepancy indicates that the lower temperature part of the anorthite phase diagram (below 1700 °C and in the pressure range of 10–20 GPa) should be more carefully investigated.

## Discussion

The well-established phase diagrams of anorthite, albite and K-feldspar suggest that these compounds decompose along normal mantle geotherm at pressures below ~3 GPa according to the following reactions:

[1] Anorthite→grossular + kyanite + quartz[10]

[2] Albite→jadeite + quartz[9]

[3] K-feldspar→$K_2Si_4O_9$-wadeite + kyanite + coesite[11]

The results of our multi-anvil experiments in conjunction with the observations of Kubo et al.[30] demonstrate that feldspars and their high-pressure derivatives can exist in the extended pressure range of up to ~15 GPa at temperatures up to ~600 °C. The persistence of feldspars to greater depths than commonly expected is also supported by recent discoveries of these materials as diamond inclusions[36–38]. Deep subduction of the crust and capture of its constituents during diamond formation has been proposed as a possible genetic process, leading to the exotic coexistence of deep mantle and crustal minerals[38–40].

Indeed, according to our results, crystalline plagioclases composing a major portion of a subducting crust may exist under the P–T conditions of (ultra)cold subducting slabs[41–43], as shown in Fig. 4. While the P–T-induced transformation pathways are well established for the average mantle geotherm (characteristic for the zones located far from continental plate boundaries), mineralogical assemblages under colder conditions where some

phase transitions are thermodynamically or kinetically retarded are still under investigation. In particular, the metastable persistence of olivine and pyroxenes has been repeatedly reported to influence the dynamics of subducting lithospheres, i.e., to affect the slab buoyancy, inhibit slab penetration in the lower mantle, and cause deep-focused earthquakes upon transformation into denser phases[44–47]. The possible influence of feldspars on slab dynamics was not considered, as these materials were believed not to withstand deep subduction. However, an effect of feldspar behaviour, particularly brittle-plastic transition, on the origin of shallow earthquakes at depths of ~10–20 km is widely discussed in the literature[48–50]. According to recent studies on the frictional properties of feldspars, these materials play a dominant role in limiting the thickness and depth of the seismogenic zone and the strength of nucleating earthquakes[22]. This is especially important for seismic events linked to deeply intruded crust, such as the enigmatic very strong New Madrid earthquake (USA, 1811–1812)[51]. Our work adds an unexpected dimension to the discussion of the possible connection between the mineral physics of rock-forming framework silicates and the nature of earthquakes. The possible persistence of metastable feldspars instead of their breakdown into denser phases may result in different seismological signatures. While the decomposition and densification of plagioclase-rich rocks is expected at ~3 GPa along the normal mantle geotherm, metastable feldspars would not contribute to seismicity until ~10 GPa (when they undergo densification through an increase in Al and/or Si coordination number). Through the whole pressure region of interest, the densities of metastable feldspars are much lower than that of the average mantle material, as shown in Fig. 3. Feldspars of certain compositions may survive in different polymorph modifications to greater depths than previously expected and may possibly contribute to intermediate- and deep-focused seismicity[52].

The behaviour of feldspars (particularly those of anorthite composition) along a subduction is of special interest for the widely discussed question of anorthositic crust fate[35,53,54]. Once formed on the Earth and its moon at ~4.5 Ga, the primordial anorthosite crust is still present on the Moon, while there is no geological evidence for its persistence on the modern Earth's surface. One hypothesis is that the Earth's primordial crust was tectonically eroded and subducted into the deep interior[55]. A recent study on the phase relations of lunar anorthosite composition in the pressure range of 12–125 GPa at ~1700 °C has shown that it has a higher density than Preliminary reference Earth model (PREM) and pyrolite in the upper mantle, while its density becomes comparable or lower under lower mantle conditions[35]. These results suggest that ancient anorthosite crust subducted down to the deep mantle was likely to be accumulated at 660–720 km in depth without coming back to the Earth's surface. Between 12 and 20 GPa, anorthosite was found to decompose into Gt + CAS + St upon heating, while Ca-Tschermak's pyroxene has not been detected, in contrast to our observations. Because of the repeatedly acknowledged metastability of pyroxene and its slow diffusion into garnet, the formation of pyroxenes may influence the density and dynamics of the descending anorthositic crust and possibly cause its stagnation at shallower depths.

Beyond the Earth's environments, the discovery of high-pressure polymorphs of feldspars and their persistence at elevated temperatures is important for planetology and exoplanetology. The abundance of framework silicates in the terrestrial-type planetary bodies indicates that their dense high-pressure polymorphs could be present in rocky interiors following moderate P–T profiles. Specifically, this finding is important for the nascent and rapidly advancing field of exoplanetology[56]. A number of discovered terrestrial planets beyond the Solar System strongly differ in size and composition from the Earth[57–59], indicating that their rock-forming minerals may follow different transformation pathways.

## Methods

**X-ray microprobe analysis of the samples.** Natural samples of anorthite (An) and albite (Ab) were provided by the Centre of Natural History (CeNak, Universität Hamburg, Hamburg, Germany). Microcline (Mi) was provided by Uppsala University, Sweden. The Ab originated from Vizze Valley (Pfitsch Valley), South Tirol, Trentino-Alto Adige, Italy, and the microcline originated from Värmland County, Sweden. The chemical compositions of the samples were characterised using wavelength-dispersive X-ray microprobe analysis (JEOL JXA-8200; focused beam; accelerating voltage of 15 keV and beam current of 15 nA (Bayerisches Geoinstitut (BGI), Bayreuth, Germany). Metallic Fe, orthoclase, albite, enstatite and spinel were used as standards for Fe, K, Na, Mg and Al, respectively. Andradite was used as a standard for Ca and Si. Atomic number effects, absorption, and fluorescence (ZAF) corrections were taken into account. The feldspar compositions in wt% of oxides are given in Supplementary Table 9. On the basis of eight oxygen atoms, chemical formulas were calculated as $Ca_{1.0}Na_{0.02}Si_{1.99}Al_{2.01}O_8$ (An), $Na_{0.98}Ca_{0.04}K_{0.01}Si_{2.91}Al_{1.09}O_8$ (Ab) and $K_{0.93}Na_{0.06}Si_{2.98}Al_{1.02}O_8$ (Mi).

**In situ high-pressure single-crystal X-ray diffraction experiments.** Three separate in situ high-pressure SCXRD were performed at the P02.2 experimental station at the Petra III synchrotron (Hamburg, Germany, experiments #1, #3, #4) and at the 13-IDD beamline at Advanced Photon Source (GSECARS, Argonne, United States, experiment #2). Symmetric diamond anvil cells (DACs) with culet diameters of 300 μm were used for pressure generation in experiment #1, while novel BX110 DACs with culet diameters of 250 μm were used in experiments #2–4 (Supplementary Note 1 and Supplementary Figs. 3 and 4). The sample chambers with approximate diameters of 130–150 μm were obtained by drilling the pre-indented rhenium gasket. Feldspar single crystals were placed inside the sample chambers along with a ruby sphere (#1) or gold particles (#2–4) for pressure characterisation[60,61]. The DACs were loaded with neon as a pressure-transmitting medium using an in-house gas loading system at Petra III and BGI[62].

Monochromatic X-ray diffraction experiments were performed using X-rays with a wavelength of ~0.29 Å. The X-ray beam was focused to <3 × 3 μm$^2$. Diffraction patterns were collected using Perkin Elmer 1621 (at Petra III) and Pilatus CdTe 1 M (at APS) detectors. Before each experiment, the detector-sample distance was calibrated with a CeO$_2$ (#1, #3, #4) and LaB$_6$ (#2) standard using the procedure implemented in the programme Dioptas[63]. Experimental details are summarised in Supplementary Table 10.

At each pressure point, both a wide-scan and a stepped ω scan were collected for each crystal. Wide-scans consisted of 40 s of exposure during rotations of ±20° of the DAC. Step scans consisted of individual exposures taken at 0.5° intervals to constrain the ω angle of maximum intensity of each peak. Collected diffraction images were analysed using the programme CrysAlis Pro© (Agilent 2012). The SHELXL programme package was used for all structural determinations[64] (Supplementary Data 1 and Supplementary Note 2). The evolution of volumes and densities along the compression is given in Supplementary Table 1, while details of representative structural refinements are given in Supplementary Tables 2–6.

**Multi-anvil experiments.** High-pressure, high-temperature experiments were performed under various P–T conditions using 1000-ton (Hymag) and 1200-ton (Sumitomo) multi-anvil hydraulic presses, as described elsewhere[65]. The Kawai-type multi-anvil system was employed. In our experiments, 14/8 (octahedron edge/anvil truncation length in mm) assemblies were used to achieve pressures of 5–17 GPa. The temperature was increased stepwise up to 300–1500 °C at a rate of ~80 °C/min. Then, the samples were quenched. After decompression, the synthesis products were extracted, and crystals were removed from the capsule with a needle and carefully washed in water. The details of the performed multi-anvil experiments are summarised in Supplementary Table 9.

**X-ray diffraction analysis of the samples recovered after multi-anvil experiments.** The recovered samples were analysed under ambient conditions at the P02.2 beamline and on an in-house diffractometer at BGI. For the experiments at P02.2, several selected pieces with an average size of ~20–60 μm were placed on a large diamond culet. The experimental setup and approach are identical to the above-described high-pressure experiments. In the case of powder samples, only wide scans were performed, while for single-crystal samples, step scans were also collected.

At BGI, larger isometric pieces of materials with linear dimensions of ~100 μm were studied using a high-brilliance Bruker diffractometer (Ag Kα radiation) equipped with an Apex Charge Coupled Device (CCD) detector.

## Data availability

The X-ray crystallographic coordinates for structures reported in this article have been deposited at the Inorganic Crystal Structure Database (ICSD) under deposition numbers CSD 1987887-1987904. These data can be obtained from CCDC's and FIZ Karlsruhe's free service for viewing and retrieving structures (https://www.ccdc.cam.ac.uk/structures/).

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

## Acknowledgements

Prof. Dr. Jochen Schlüter (Mineralogisches Museum, Centrum für Naturkunde (CeNak), Universität Hamburg, Grindelallee 48, 20146 Hamburg) is highly acknowledged for providing the samples. L.D. acknowledges financial support provided by DFG grants DU 393/9-2 and DU 393/13-1. L.G. was supported by the Russian Science Foundation, grant number 19-77-00038. This research was carried out at the light source PETRA III at DESY, a member of the Helmholtz Association (HGF), and at GeoSoilEnviroCARS (Sector 13), Advanced Photon Source (APS), Argonne National Laboratory. GeoSoilEnviroCARS is supported by the National Science Foundation-Earth Sciences (EAR-1634415) and Department of Energy-Geosciences (DE-FG02-94ER14466). Use of the Advanced Photon Source was supported by the U.S. Department of Energy, Office of Science, Office of Basic Energy Sciences, under Contract No. DE-AC02-06CH11357.

## Author contributions

A.P., I.K., E.K., G.A., M.B., L.G., T.F. and V.P. conducted the high-pressure single-crystal X-ray diffraction experiments. D.S. and I.K. performed ex situ multi-anvil experiments. A.P. analysed the X-ray diffraction data. A.P. and L.D. interpreted the results and wrote the paper with the contribution of all authors.

## Competing interests

The authors declare no competing interests.
