## [Peer Review File · Nature Communications]

Reviewers' comments:

Reviewer #1 (Remarks to the Author):

The quality of the crystallographic data is quite impressive, and the authors are to be congratulated on their successful efforts to obtain such data. However, I am disappointed that much of the data collected (Supplementary Table 1) is not reported in full, nor used in this manuscript. I am also disappointed that the authors do not discuss the mechanisms of the coordination changes at high pressure, and whether they are related to previously reported elastic softening and recovery in feldspars at high pressure. Were all of the data to be reported and analysed, this would make an excellent contribution.

However, I disagree fundamentally with the two major interpretations by the authors of the results reported here:

(1) One of the precepts of this manuscript is, as stated in the abstract, that 'Previous structural studies of feldspars are limited to ~10 GPa, and it is commonly believed that their pressure-induced deformations are realized through tilting of AlO_4 and SiO_4 tetrahedra in a tetrahedral framework'. First, belief is for religion not science. Second, this is not true. For example, figure 4 in reference 22 (from 1996) shows the experimentally-measured compression of the tetrahedra in microcline up to 7 GPa, and the DFT simulations of reference 13 show the compression of the tetrahedra in albite. Curetti et al (2011) *Amer Min*, 96, 383-392, also measure tetrahedral compression in a feldspar. Thus, the amount of compression of the tetrahedra is small and therefore difficult to measure experimentally, also because the suppression of the anisotropic thermal motion of the bridging oxygens masks some of the compression as measured by Bragg diffraction. But the shift under pressure of Raman lines due to internal modes of the tetrahedra also indicates that there is tetrahedral compression. Therefore, as stated in many studies, and stated here on page 3 'the pressure-induced compression of the framework is primarily accommodated by altering the TO- T bond angles' but nobody 'believes' that the tetrahedra are rigid. The use of models with rigid tetrahedra, which dates back to Megaw in the 1960s is only to help understand the role of tetrahedral rotation in the structural response of feldspars and to identify the role of tetrahedral distortions and compression.

Therefore the abstract needs correction. And the following phrase on page 3 'while the TO_4 tetrahedra remain as rigid units' is false and must be deleted.

The text at the bottom of page 5 and the top of page 6 also has this mistake by implying previous studies did not detect tetrahedral compression. Then the new structural phase transitions are discussed in terms of a 'compression mechanism'. I think that it is wrong to consider transition mechanisms as the same type of physical process as that of phase transformations and to do so will only confuse the reader; one would not consider polymorphic phase transitions (e.g. from andalusite to sillimanite) as a change in compression mechanism. I would therefore recommend re-writing the text in page 6 to avoid this confusion.

(2) I am not convinced that the heating experiments demonstrate conclusively that feldspars could remain stable at pressures above 3 GPa in subducting slabs on geological timescales, as the experiments only lasted 2 hours. One also imagines that the kinetics of transformations to the stable phases would be greatly accelerated in the presence of water: were water-present experiments performed?

I also have a number of technical questions and comments:

What is the evidence for the space groups reported for these structures? Could they actually be acentric, and what statistical tests were performed to test this possibility?

Were single crystals preserved through the transitions? If not, what procedures were used to separate the diffraction contributions from individual grains? Were they related by twin operations, etc?

Supplementary Table 1 indicates that data were collected at many more pressures than those discussed in the manuscript, which focusses on the new polymorphs.

- Does the evolution of the unit cell parameters (not reported) agree with previous measurements below 10 GPa?

- Were full intensity datasets collected at these lower pressures? If so, do the refined structure agree with previous determinations or, if not, what are the differences?
- If full structure refinements were performed on these other datasets, they would be extremely valuable for understanding the mechanisms of structural compression of framework materials and should be published in full.
- Did the authors detect the elastic softening in albite (references 13 and 20)? Can they shed light on the mechanism of the elastic recovery above 9 GPa? Is the coordination change they report the consequence of this softening and the reason for recovery? Is it already present in the DFT simulations of ref 13? Is the coordination change the reason for the changes in the Raman spectra in the 8 – 11 GPa pressure range (Aliatis et al (2016) Phys Chem Minerals, doi: 10.1007/s00269-016-0850-5)?
- What is the criteria for determining whether an atom-atom distance constitutes a bond? Bond valence sums, or some other method?
- What is the mechanism of the transformations involving coordination change, and is there hysteresis? This could be addressed if there were more closely-spaced data points around the transition pressures. Is the mechanism the same for the coordination change for Si in CaSi₂O₅ which occurs around 1 kbar, or in the structures described in references 25-27? What are the general features of this type of coordination change?

Other corrections:

Figure 1: This shows well the topology of the structure, but the unit-cell of a feldspar depends on the Al:Si ratio and the state of Al/Si order. While the differences in the alpha and gamma angles between the monoclinic and triclinic polymorphs is probably not significant at the scale of the drawing, the doubling of the c-axis in anorthite compared to anorthite would be obvious. Therefore, the statement in the figure caption 'Black lines outline the unit cell' is misleading. Either say the 'unit-cell of alkali feldspars' or 'of the aristotype structure', or reword in another way.

Page 3, line 1: 'Commonly, feldspars are believed to be stable at pressures...'. Again, belief is for religions not science. Better to state that 'Feldspars are thermodynamically stable at pressures up to 3 GPa' The highest thermodynamic stability is for K-feldspar, at about 3 GPa.

Reference list: requires copy-editing. Many chemical formula are given without subscripts. Some journal names are not written correctly, etc

Reviewer #2 (Remarks to the Author):

It is not often that new polymorphs of the most important rock forming minerals are discovered, so this paper will certainly be of interest across a wide area of mineralogy, petrology and geophysics. The authors have demonstrated that each of the three principal end-members of the ternary feldspar system undergo phase transitions to previously unknown high pressure phases. Of particular interest and importance is the increase in coordination of Al and Si from [4], typical of the crust, to [5] or [6]. Whether this turns out to be important in the context of the properties of the crust at depth remains to be seen, because these minerals usually react with other minerals before pressures of 14-15 GPa are reached. That does not reduce the impact of the work, however. It is likely that there will be a whole family of structures with compositions comparable with the many different natural and synthetic feldspars that are known.

The paper is clear and well written. The quality of the data appears to be good from what was probably quite a difficult series of experiments, from a technical point of view.

In my opinion, this is the kind of work that should appear in Nature journals to broaden the rather narrow range of geological papers that are currently published.

Reviewer #1 (Remarks to the Author):

>> We are thankful to the Reviewer#1 for the thoughtful reading of our manuscript and his/her valuable comments and questions. We have modified the manuscript and reported data accordingly. Please find below point-by-point answers to the raised questions. The modifications in text and supplementary materials are marked with yellow.

The quality of the crystallographic data is quite impressive, and the authors are to be congratulated on their successful efforts to obtain such data. However, I am disappointed that much of the data collected (Supplementary Table 1) is not reported in full, nor used in this manuscript.

>> In the first version of the manuscript we have mainly focused on the new polymorphs featuring the increased coordination number of Al and Si atoms and on geological implications of these discoveries. However, we agree with the Reviewer that lower pressure data illuminates the mechanism of coordination number increase and is of general importance for high-pressure crystal chemistry of feldspars. Therefore, we have extended materials reported in the present article. The Supplementary Table 1 includes now unit cell parameters at all studied pressure points. In addition, we report the crystal structure refinements at pressures extending the previous studies: above 8 GPa for anorthite, above 10 GPa for albite and above 9 GPa for microcline. These CIFs will be deposited to the ICSD as well. Most of the new data are discussed in the manuscript. In our opinion, data of microcline softening prior to the phase transition is of particular interest as were not reported before.

I am also disappointed that the authors do not discuss the mechanisms of the coordination changes at high pressure, and whether they are related to previously reported elastic softening and recovery in feldspars at high pressure. Were all of the data to be reported and analysed, this would make an excellent contribution.

>> We have added a paragraph describing the mechanism of coordination number changes and discussed its relation to the previously reported elastic softening as well. We believe that pressure-induced geometrical distortion are indeed correlated with the elastic softening, as, for example, it is demonstrated by behaviour of microcline in the pressure region 9-10.5 GPa. Regarding anorthite, additional structural data between ~7-11 GPa would be needed to further clarify this question. This will become subject of our upcoming studies.

However, I disagree fundamentally with the two major interpretations by the authors of the results reported here:

(1) One of the precepts of this manuscript is, as stated in the abstract, that 'Previous structural studies of feldspars are limited to ~10 GPa, and it is commonly believed that their pressure-induced deformations are realized through tilting of AlO₄ and SiO₄ tetrahedra in a tetrahedral framework'. First, belief is for religion not science. Second, this is not true. For example, figure 4 in reference 22 (from 1996) shows the experimentally-measured compression of the tetrahedra in microcline up to 7 GPa, and the DFT simulations of reference 13 show the compression of the tetrahedra in albite. Curetti et al (2011) *Amer Min*, 96, 383-392, also

measure tetrahedral compression in a feldspar. Thus, the amount of compression of the tetrahedra is small and therefore difficult to measure experimentally, also because the suppression of the anisotropic thermal motion of the bridging oxygens masks some of the compression as measured by Bragg diffraction. But the shift under pressure of Raman lines due to internal modes of the tetrahedra also indicates that there is tetrahedral compression. Therefore, as stated in many studies, and stated here on page 3 ‘the pressure-induced compression of the framework is primarily accommodated by altering the TO- T bond angles’ but nobody ‘believes’ that the tetrahedra are rigid. The use of models with rigid tetrahedra, which dates back to Megaw in the 1960s is only to help understand the role of tetrahedral rotation in the structural response of feldspars and to identify the role of tetrahedral distortions and compression.

Therefore the abstract needs correction. And the following phrase on page 3 ‘while the TO4 tetrahedra remain as rigid units’ is false and must be deleted.

The text at the bottom of page 5 and the top of page 6 also has this mistake by implying previous studies did not detect tetrahedral compression. Then the new structural phase transitions are discussed in terms of a ‘compression mechanism’. I think that it is wrong to consider transition mechanisms as the same type of physical process as that of phase transformations and to do so will only confuse the reader; one would not consider polymorphic phase transitions (e.g. from andalusite to sillimanite) as a change in compression mechanism. I would therefore recommend re-writing the text in page 6 to avoid this confusion.

> > We completely agree with the Reviewer that TO₄ tetrahedra undergo slight compression below 10 GPa and that usage of term “rigid” was misleading. Following the recommendations, we have modified the text. We also convinced by Reviewer that that phase transitions should not be considered as a “compression mechanism”. Accordingly, we have modified this part of the Discussion as well.

(2) I am not convinced that the heating experiments demonstrate conclusively that feldspars could remain stable at pressures above 3 GPa in subducting slabs on geological timescales, as the experiments only lasted 2 hours. One also imagines that the kinetics of transformations to the stable phases would be greatly accelerated in the presence of water: were water-present experiments performed?

>> Indeed, time is the likely to be the only parameter in modern high-pressure techniques that can not be extended up to the scales of nature. In our studies, we have heated for 2 hours while the previous *in situ* multi anvil experiments (that also shown HT-HP annealing of the feldspars) have been limited to timescales of the several minutes (Kubo et al., 2010). Agreement of our results independently on the heating time scale hints that the same results would be expected for longer heating. We have not performed the dedicative water-rich multi anvil experiments. However, as soon as we have used non-dehydrated NaCl as a pressure medium, we assume slightly humid conditions of our HT-HP experiments. Inspired by Reviewer comment we performed additional experiment by heating of anorthite at 5 GPa in wet NaCl during 110 h at 500

°C; we found that anorthite survived. Moreover, the discoveries of the feldspars in diamond inclusion directly indicate that there is a mechanism to maintain feldspars existence in the deep Earth's interior. In our work, we undertook an attempt to propose one of the possible geological scenarios, and, undoubtedly, more studies are needed to fully resolve this question.

I also have a number of technical questions and comments:

What is the evidence for the space groups reported for these structures? Could they actually be acentric, and what statistical tests were performed to test this possibility?

>> We have performed Wilson's statistical and E-statistics tests (implemented in the work package WinGX). The Wilson plots indicate that the new phases are centrosymmetric with probabilities of 89% for microcline-II, 88% for anorthite-III, 94% and 91% for albite-II and -III, respectively. $|E^2-1|$ criterion is of 1.001 for microcline-II, 0.99 for anorthite-III, 0.981 and 0.980 for albite-II and -III, respectively, also strongly indicating that the structures are centrosymmetric.

Were single crystals preserved through the transitions? If not, what procedures were used to separate the diffraction contributions from individual grains? Were they related by twin operations, etc?

>> For all three compositions, anorthite, microcline and albite, the single crystal nature of the sample has been preserved across the phase transitions, both upon compression and decompression (in case of albite).

Supplementary Table 1 indicates that data were collected at many more pressures than those discussed in the manuscript, which focusses on the new polymorphs.

- Does the evolution of the unit cell parameters (not reported) agree with previous measurements below 10 GPa?

>> In revised manuscript, we report the evolution of the unit cell parameters in Supplementary Table 1 and Supplementary Figure 1. Our data are in good agreement with the previous measurements. Small mismatches can be related to slightly different chemical compositions or/and differences in experimental procedures.

- Were full intensity datasets collected at these lower pressures? If so, do the refined structure agree with previous determinations or, if not, what are the differences?

>> Full datasets were collected at the all pressure points reported in Supplementary Table 1. At low pressures, the refined structures only negligible differ from the ones reported earlier. In the re-submission we attach all CIFs for the crystal structures in the extended pressure ranges: above 8 GPa for anorthite, above 10 GPa for albite and above 9 GPa for microcline. These CIFs will be deposited to the ICSD as well.

- If full structure refinements were performed on these other datasets, they would be extremely valuable for understanding the mechanisms of structural compression of framework materials and should be published in full.

>> We have extended the Supplementary Materials of the article and now we report full crystal structure refinements at points extending previously studied pressure ranges: above 8 GPa for anorthite, above 10 GPa for albite and above 9 GPa for microcline. These CIFs will be deposited to the ICSD as well.

- Did the authors detect the elastic softening in albite (references 13 and 20)? Can they shed light on the mechanism of the elastic recovery above 9 GPa? Is the coordination change they report the consequence of this softening and the reason for recovery? Is it already present in the DFT simulations of ref 13? Is the coordination change the reason for the changes in the Raman spectra in the 8 – 11 GPa pressure range (Aliatis et al (2016) Phys Chem Minerals, doi: 10.1007/s00269-016-0850-5)?

>> Yes, our data on albite prior to the phase transition are in good agreement with previously reported data and confirm the presence of elastic softening. We have measured the only one data point above 9 GPa (*i.e.*, at 11.5 GPa) that is insufficient to deeply discuss the mechanism of the elastic recovery. However, our results show that the initial tetrahedral framework of albite is preserved at least up to 11.5 GPa. It indicates that the recently reported anomalous elastic behaviour (Aliatis et al., 2016; Mookherjee et al., 2016) are not caused by the increased coordination number of T atoms. However, it can be correlated with the geometrical distortion of AlO_4 tetrahedron, as obvious in the case of microcline.

- What is the criteria for determining whether an atom-atom distance constitutes a bond? Bond valence sums, or some other method?

>> Firstly, whether a phase transition is induced by progressive approach of additional oxygen O^* into coordination sphere of T atom and is accompanied by well-pronounced jump in unit cell parameters, it indicates that formation of a T- O^* bond has occurred. From the previous studies, a T-O contact shorter than 2 Å have been shown to constitute a bond. Then, for the contacts longer than 2 Å, T-O bond distributions have been analyzed. The small differences between T-O contacts below and above 2 Å (~0.1-0.3 Å) likely indicates that T-O should be considered as a bond. According to this logic, for instance, we have called “bonds” all contacts of Al^7 (1.740, 1.745, 1.77, 1.907, 2.156, 2.16 Å) in anorthite-III. Meanwhile, the coordination of Al^4 in the same crystal structure with bond contacts of 1.727, 1.749, 1.783, 1.78, 2.22 Å we have called [4+1]. We are aware that this qualitative approach does not provide a strict answer to the question on T-O bonding. However, we are convinced that it is efficient enough for the description of the topologies of the new polymorphs.

- What is the mechanism of the transformations involving coordination change, and is there hysteresis? This could be addressed if there were more closely-spaced data points around the transition pressures. Is the mechanism the same for the coordination change for Si in CaSi_2O_5

which occurs around 1 kbar, or in the structures described in references 25-27? What are the general features of this type of coordination change?

>> We think that the coordination change is induced by progressive geometrical distortion of TO₄ tetrahedron along the densification of the framework and continuous approach of an additional oxygen atom into coordination sphere of the T atom. Unfortunately, the decompression data is available only for albite and sparse (Supplementary Table 1) that does not allow us to estimate the hysteresis. In future, we intend to investigate this question in detail. The same mechanism of transformation we have observed in our previous studies (ref. 25-27) of frameworks with different topology. Among general features of this type of coordination change, we can note:

- * geometrical distortion of TO₄ tetrahedra (*i.e.* displacement of the T atom from the center of tetrahedron towards the O* oxygen);

- * such distortion is often accompanied by anomalous behaviour of unit cell parameters;

- * in most of the cases the final formation of T-O* bond is accompanied by a well-pronounced jump in unit-cell parameters. However, the sluggish transitions are also possible.

- * newly formed TO₅ polyhedra are of either trigonal-bipyramidal or of square pyramidal geometries. However, it is likely that square pyramidal geometry is a transitional step in the formation of octahedra, as indicated by the progressive approach of a sixth oxygen in albite-III. Meanwhile, the newly TO₆ polyhedra are exclusively of octahedral geometry.

We have added discussion on this type of the coordination change into the manuscript.

Other corrections:

Figure 1: This shows well the topology of the structure, but the unit-cell of a feldspar depends on the Al:Si ratio and the state of Al/Si order. While the differences in the alpha and gamma angles between the monoclinic and triclinic polymorphs is probably not significant at the scale of the drawing, the doubling of the c-axis in anorthite compared to anorthite would be obvious. Therefore, the statement in the figure caption 'Black lines outline the unit cell' is misleading. Either say the 'unit-cell of alkali feldspars' or 'of the aristotype structure', or reword in another way.

>> Corrected.

Page 3, line 1: 'Commonly, feldspars are believed to be stable at pressures...'. Again, belief is for religions not science. Better to state that 'Feldspars are thermodynamically stable at pressures up to 3 GPa' The highest thermodynamic stability is for K-feldspar, at about 3 GPa.

>> Corrected.

Reference list: requires copy-editing. Many chemical formula are given without subscripts. Some journal names are not written correctly, etc

>> Corrected.

REVIEWERS' COMMENTS:

Reviewer #1 (Remarks to the Author):

The revision addresses most of the criticisms that I had of the original submission, and the addition of the crystallographic data strengthens the arguments made by the authors in their manuscript.

I still think that the criteria used to define what constitutes a bond, and thus a change in coordination number, is somewhat arbitrary, but the revised manuscript does at least make clear what criteria have been used and the deposited data allow others to make their own analysis.

One new request. On closer inspection of the data, I realise that the authors have used for the new phases a choice of unit-cell orientation that is not obviously related to the conventional unit-cell setting of feldspars. It would be very useful if they could provide the transformation matrices between the conventional cell setting and the settings used for these high-pressure polymorphs. The authors can determine these unambiguously from the orientation matrices used to index their diffraction patterns, and they should be added to the supplementary materials.

With respect to the relative rigidity of the tetrahedra as determined in previous studies, the text is still not exactly correct at several points, and I would suggest the following specific rewording at various points:

Line 17-19: Still poorly written and misleading. I would suggest '...to ca. 10 GPa, and have shown that the dominant mechanism of pressure-induced deformation is the tilting of....'

Line 60: Still incorrect. The tetrahedral compression is not 'insignificant'. That would imply that it is less than the experimental uncertainties and therefore cannot be determined. However, that is not true, as tetrahedral compression has been measured experimentally (just above the levels of uncertainty) and is very clear in DFT simulations. Therefore, rewrite as 'while the TO4 tetrahedra show very little compression and behave as relatively-rigid units.'

Line 186: 'while the tetrahedra undergo little distortion and compression'.

Line 188: I would leave in the original text 'a different mechanism, i.e.'

I am not sure now that the argument on lines 190-194 about ionic potential and coordination change are correct, as one might expect that the ability to change coordination is actually related to the energy change involved in the change of orbital hybridisation. But I would leave in the current text, as it might provoke discussion of this issue.

Minor corrections:

Line 105: Change to 'by the absence of single crystal reflections from....'

Line 134: Change to 'bond length'

Line 140: Cut and paste error? I think this should be 'albite-III' not 'albite-II'

Line 151: geometry

Line 189: hyphenate 'nearly-rigid' as a compound adjective. ...the rule is that if you use two words

as a compound adjective, you must hyphenate them. But not if the words are used as an adjective plus noun. So, 'high-pressure structure'(compound adjective) and 'a structure at high pressure' (adjective 'high', plus noun 'pressure')

Line 190: delete 'soon'

Line 215: change to: 'insufficient to draw conclusions about pre-transitional...'

Line 216: change to: 'However, increases in the distortion parameters of the AlO₄...'

Line 220 'connectivities' if the referenced structures are of more than one topology

Line 220-221: change to 'Similar to feldspars, ...'

Line 221: 'continuous'

Line 223 hyphenate 'newly-formed'

Line 229: delete 'well'; you cannot qualify an absolute adjective like 'pronounced'. Something is either 'pronounced' or it is not! I would also rewrite this part: 'a pronounced jump in the unit cell parameters, as also found in the current study and, accordingly, an increase in the densities of the feldspar-derived phases (Figure 4).'

Line 232: hyphenate 'previously-reported'

Line 255: 'the stability'

Line 341-343. Rewrite. This sentence makes no sense to me. Same for lines 357-359; what is 'Earth's paradigm'?

Ross Angel

IGG-CNR Padova, 13-March-2020

Rebuttal Letter

Reviewer #1 (Remarks to the Author):

The revision addresses most of the criticisms that I had of the original submission, and the addition of the crystallographic data strengthens the arguments made by the authors in their manuscript.

I still think that the criteria used to define what constitutes a bond, and thus a change in coordination number, is somewhat arbitrary, but the revised manuscript does at least make clear what criteria have been used and the deposited data allow others to make their own analysis.

One new request. On closer inspection of the data, I realise that the authors have used for the new phases a choice of unit-cell orientation that is not obviously related to the conventional unit-cell setting of feldspars. It would be very useful if they could provide the transformation matrices between the conventional cell setting and the settings used for these high-pressure polymorphs. The authors can determine these unambiguously from the orientation matrices used to index their diffraction patterns, and they should be added to the supplementary materials.

>> Added.

With respect to the relative rigidity of the tetrahedra as determined in previous studies, the text is still not exactly correct at several points, and I would suggest the following specific rewording at various points:

Line 17-19: Still poorly written and misleading. I would suggest ‘...to ca. 10 GPa, and have shown that the dominant mechanism of pressure-induced deformation is the tilting of...’

>> Corrected.

Line 60: Still incorrect. The tetrahedral compression is not ‘insignificant’. That would imply that it is less than the experimental uncertainties and therefore cannot be determined. However, that is not true, as tetrahedral compression has been measured experimentally (just above the levels of uncertainty) and is very clear in DFT simulations. Therefore, rewrite as ‘while the TO4 tetrahedra show very little compression and behave as relatively-rigid units.’

>> Corrected.

Line 186: ‘while the tetrahedra undergo little distortion and compression’.

>> Corrected.

Line 188: I would leave in the original text ‘a different mechanism, i.e.’

>> Corrected.

I am not sure now that the argument on lines 190-194 about ionic potential and coordination change are correct, as one might expect that the ability to change coordination is actually related to the energy change involved in the change of orbital hybridisation. But I would leave in the current text, as it might provoke discussion of this issue.

Minor corrections:

Line 105: Change to 'by the absence of single crystal reflections from...'

>> Corrected.

Line 134: Change to 'bond length'

>> Corrected.

Line 140: Cut and paste error? I think this should be 'albite-III' not 'albite-II'

>> Corrected.

Line 151: geometry

>> Corrected.

Line 189: hyphenate 'nearly-rigid' as a compound adjective. ...the rule is that if you use two words as a compound adjective, you must hyphenate them. But not if the words are used as an adjective plus noun. So, 'high-pressure structure' (compound adjective) and 'a structure at high pressure' (adjective 'high', plus noun 'pressure')

>> Corrected.

Line 190: delete 'soon'

>> Corrected.

Line 215: change to: 'insufficient to draw conclusions about pre-transitional...'

>> Corrected.

Line 216: change to: 'However, increases in the distortion parameters of the AlO₄...'

>> Corrected.

Line 220 'connectivities' if the referenced structures are of more than one topology

>> Corrected.

Line 220-221: change to 'Similar to feldspars, ...'

>> Corrected.

Line 221: 'continuous'

>> Corrected.

Line 223 hyphenate 'newly-formed'

>> Corrected.

Line 229: delete 'well'; you cannot qualify an absolute adjective like 'pronounced'.
Something is either 'pronounced' or it is not! I would also rewrite this part: 'a pronounced jump in the unit cell parameters, as also found in the current study and, accordingly, an increase in the densities of the feldspar-derived phases (Figure 4).'

>> Corrected.

Line 232: hyphenate 'previously-reported'

>> Corrected.

Line 255: 'the stability'

>> Corrected.

Line 341-343. Rewrite. This sentence makes no sense to me. Same for lines 357-359; what is 'Earth's paradigm'?

>> Corrected.